# WHEN IN DOUBT, SUMMON THE TITANS: EFFICIENT INFERENCE WITH LARGE MODELS

## ABSTRACT

Scaling neural networks to "large" sizes, with billions of parameters, has been shown to yield impressive results on many challenging problems. However, the inference cost incurred by such large models often prevent their application in most real-world settings. In this paper, we propose a two-stage framework based on distillation that realizes the *modelling* benefits of the large models, while largely preserving the *computational* benefits of inference with more lightweight models. In a nutshell, we use the large teacher models to guide the lightweight student models to only make correct predictions on a subset of "easy" examples; for the "hard" examples, we fall-back to the teacher. Such an approach allows us to efficiently employ large models in practical scenarios where easy examples are much more frequent than rare hard examples. Our proposed use of distillation to only handle easy instances allows for a more aggressive trade-off in the student size, thereby reducing the amortized cost of inference and achieving better accuracy than standard distillation. Empirically, we demonstrate the benefits of our approach on both image classification and natural language processing benchmarks.

## 1 INTRODUCTION

Scaling neural networks to "large" sizes has brought dramatic quality gains over a wide variety of machine learning problems, including at the tails. In computer vision, the high performing models for image classification (Kolesnikov et al., 2019; Xie et al., 2020; Tan & Le, 2019; Foret et al., 2021) and segmentation (Ghiasi et al., 2020) have upto 928M parameters and require up to 600G FLOPs for a prediction. Similarly, in natural language processing, transformer-based approaches, which have several billion parameters and require up to a tera-FLOP for a prediction, are leading performance on language understanding tasks (Raffel et al., 2019; Brown et al., 2020; Fedus et al.) and neural machine translation (Bapna & Firat, 2019; Huang et al., 2018).

The immensely expensive inference cost of these large models is, however, hindering their direct widespread adoption (Jouppi et al., 2017; Ning, 2013; Crankshaw et al., 2017; Zhang et al., 2019). The issue is further exacerbated in deployment over resource-constrained edge devices such as mobile phones (Zhang et al., 2020). As a workaround, many model compression techniques have been proposed to reduce the computational cost and memory footprint by trading-off accuracy, including quantization (Mozer & Smolensky, 1988; Han et al., 2015), pruning (LeCun et al., 1989; Hassibi & Stork, 1993), and distillation (Bucilă et al., 2006; Romero et al., 2014; Hinton et al., 2015). However, there is a limit to how far such model compression techniques can be pushed to reduce inference cost while retaining good performance across all inputs (cf. teacher-student accuracy gaps in (Cho & Hariharan, 2019; Menon et al., 2020b; Mirzadeh et al., 2020; Wang et al., 2017a)).

Ideally, the compute required to make predictions on an instance should depend on the hardness of the instance. But the large models do not adapt their computational budget based on the complexity of the task at hand. We conjecture that the full ability of a large model is needed only for a small fraction of "hard" instances. The majority of real-inputs are "easy", for which performing full computation of a large model is wasteful; rendering the overall ML system inefficient. Such an inefficient utilization of compute gets even more pronounced for many real-world data that are heavily long-tailed (Zhu et al., 2014; Wang et al., 2017b; Van Horn & Perona, 2017), with hard instances belonging to the tail.

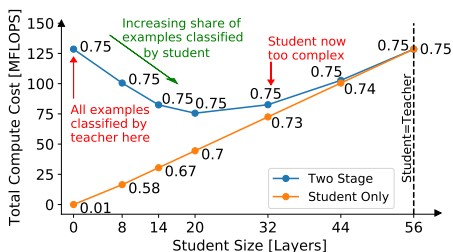

Figure 1: **Left:** A schematic of the proposed two-stage inference framework where after distillation we retain both the lightweight student model and the large teacher model. At inference time if the student finds an instance hard, we fall back to the teacher for a prediction. **Right:** The two-stage framework on CIFAR-100 image classification task using ResNets allows us to aggressively trade-off size of the student, thereby reducing the overall computation in expectation and achieving better accuracy compared to performing inference based on only the student. Note that the numbers annotated at each (student size, total compute cost) point on the plots denote the overall accuracy of the corresponding setup. For the two-stage inference, as we increase the size of the student, we can always achieve an accuracy of 0.75 by delegating an appropriate fraction of instances to the teacher. Compared to classifying all examples using the teacher, computation cost savings come from those instances where the student makes the final prediction.

In this paper, we focus on realizing the benefits of a large model on the hard instance without incurring the unnecessary large inference cost on prevalent easy instances. Towards this, we propose to employ a novel *distillation-based two-stage inference* framework in Figure 1 (left): First use a lightweight student model to make a prediction. If the student is confident, we emit the prediction and we want the student to be confident on all the easy instances, which should be a large fraction of the test time queries. When the student is in doubt, ideally only for a small number of hard examples, we fall-back to the large teacher. Our main contributions for leveraging the excellent performance of large models to realize a desirable *inference cost vs. performance* trade-off are as follows.

- The instance-aware two-stage inference mechanism crucially relies on the ability of the student model to detect the "hardness" of an input instance on the fly and routing it to the large model. To enable this routing, we propose modified distillation procedures. In particular, we employ novel distillation loss functions (cf. Sec. 4) such that the student gets penalized heavily for making mistakes on easy examples while for harder out-of-domain examples we encourage the student to be less confident, e.g., the prediction distribution be closer to the uniform distribution.

- We conduct a detailed empirical evaluation of the proposed distillation-based two-stage inference framework (cf. Sec. 5) and show that it allows us to much more aggressively trade-off size of the student for multiple image classification and natural language processing (NLP) benchmarks. Interestingly, as summarized in Figure 1, there is a sweet spot where we can achieve the same accuracy as the teacher with 45% less compute. This benefit is further magnified when considering only in-domain examples. Thus, we can reduce the overall computation over the data distribution and achieve better accuracy than performing inference with only the student model.

Note that, traditionally, the distillation approach aims to utilize a complex model to learn a simple model that has its overall performance as close to the complex model as possible. This is done under the assumption that during the inference time one can 'throw away' the complex model and rely on only the simple model for the final predictions. We would like to highlight that our goal is *not* to train a student/simple model that will be used as a standalone model to generate predictions.

It's worth mentioning that the proposed two-stage inference can also be useful in a modern setup like edge computing and 5G cloudlets (Fang et al., 2019), where a lightweight student model runs on a device to make most of the predictions with low latency and only once in a while a hard instance is delegated to a shared large teacher model running in the cloud.

## 2 RELATED WORK

Techniques to reduce inference cost for deep models mainly fall under two different approaches: quantization and pruning, and adaptive computation.

**Quantization and pruning.** The primary way suggested in the literature to accelerate predictions from deep neural networks has been quantization and pruning (Mozer & Smolensky, 1988; LeCun et al., 1989; Hassibi & Stork, 1993; Li et al., 2020; Carreira-Perpinán, 2017; Howard et al., 2019). Significant progress was made by introducing Huffman encoding methods for non-uniform quantization which led to a reduction in network sizes by orders of magnitude and up to 4x reduction in overall prediction cost (Han et al., 2015). Since then pruning and quantization have been widely adopted in computer vision and more details can be found in the recent survey by Liang et al. (2021). In the NLP domain, Gordon et al. (2020); Zadeh et al. (2020) proposed pruning BERT during training, which resulted in 30%-40% reduction in model size with minimal effect on the accuracy of the final task, however, not much compute/time savings was observed as arbitrary sparsity might not be leveraged by modern hardware accelerators. Towards this, structured pruning is more beneficial, as it removes a series of weights that correspond to an entire component of the model (Ganesh et al., 2020). In transformers, this would correspond to pruning out entire attention heads (Kovaleva et al., 2019; Raganato et al., 2020) or encoder units (Fan et al., 2019). Our proposed approach of two-stage inference is complementary to such techniques and can be combined with these to further reduce the inference cost.

**Adaptive computation.** In line with our proposed approach, there have been works trying to adapt the amount of computation of neural model based on an input instance. Effort in this space started in the vision community for enabling real-time object detection by Rowley et al. (1998) and later formalized by Viola & Jones (2001). The basic idea was to design a cascade of independent classifier and reject early on and cheaply. The idea has been generalized from a linear chain of cascaded classifiers to trees (Xu et al., 2014). Instead of combining many independent classifiers, a similar idea to stop early has emerged in monolithic deep models. One approach to this problem is represented by Adaptive Computation Time (ACT) (Graves, 2016; Chung et al., 2016). ACT is a mechanism for learning a scalar halting probability, called the "ponder time", to dynamically modulate the number of computational steps needed for each input. An alternative approach is represented by Adaptive Early Exit Networks (Bolukbasi et al., 2017), which gives the network the ability to exit prematurely - i.e., not computing the whole hierarchy of layers - if no more computation is needed. A modern incarnation of this approach in NLP with transformers encoders appeared in Schwartz et al. (2020); Liu et al. (2020); Dabre et al. (2020). This idea has been extended to generative tasks as well, where a number of decoder layers per time step are adapted in Elbayad et al. (2020). As a further generalization, Bapna et al. (2020) introduced "control symbols" to determine which components are skipped in a transformer, i.e. not all previous components need to be executed. Similar ideas had already existed in the vision community, for example, Wang et al. (2017a; 2018) introduced a method for dynamically skipping convolutional layers. All of these approaches are specialized to a task and rely on designing the whole pipeline from scratch which can be expensive if we want to achieve state-of-the-art (SoTA) performance. In contrast, we want to design efficient inference techniques achieving SoTA performance by only training cheap components, like the student model using a novel distillation procedure, while leveraging existing SoTA large models without re-training or modifying them. Moreover, our approach is a generic framework to leverage the large models independent of the underlying model architecture and problem domain. Our proposed approach ensures that large model is only invoked on instances that necessarily benefit from its large model capacity and a lite distilled model suffices to predict a large portion of test instances.

## 3 BACKGROUND

### 3.1 MULTICLASS CLASSIFICATION

Consider a standard multiclass classification problem where given an instance $x \in \mathcal{X}$, the objective is to classify the instance as a member of one of the $L$ classes, indexed by $\mathcal{Y} \triangleq [L]$. In the most common setting, given a training set comprising of $n$ instance and label pairs or training examples $S_n = \{(x_i, y_i)\}_{i \in [n]}$, one learns a classification model $f : \mathcal{X} \rightarrow \mathbb{R}^L$, where $f(x) = (f(x)_1, \ldots, f(x)_L)$ represent the model scores assigned to instance $x$ for $L$ classes. Based on the model scores, an instance can be predicted to belong to class $\hat{y}_x \triangleq \arg\max_{i \in [L]} f(x)_i$. Accordingly, the model (top-1) accuracy is defined as

$$\mathbb{P}[\hat{y}_x = y] = \mathbb{E}_{X,Y}[\mathbb{1}_{\hat{y}_x = y}] = 1 - \mathbb{E}_{X,Y}[\mathbb{1}_{\hat{y}_x \neq y}]. \tag{1}$$

Ideally, one prefers a classification model with a high accuracy. Let $\ell : \mathcal{Y} \times \mathbb{R}^L \to \mathbb{R}$ be a surrogate loss function such that $\ell(y, f(x))$ closely approximates the misclassification error $\mathbb{1}_{\hat{y}_x \neq y}$. Typically, given the training set $S_n$ and the loss function $\ell$, one selects a desired classification model via empirical risk minimization (ERM). *Softmax cross-entropy loss* is one of the most widely used surrogate loss functions for multiclass classification: Given model scores $f(x) = (f(x)_1, \ldots, f(x)_L)$, one computes the *softmax distribution*

$$\hat{p}_{f,x}(i) = e^{\tau \cdot f(x)_i} / \sum\nolimits_{j \in [L]} e^{\tau \cdot f(x)_j}, \ \ \text{for } y \in [L], \tag{2}$$

where $\tau$ denotes the temperature parameter of the softmax operation.[1] Further, we define $p_y \in \{0,1\}^L$ to be the *one-hot label distribution* corresponding to the true label $y \in [L]$, which has non-zero value at only $y$-th coordinate. Now, softmax cross-entropy loss corresponds to the distance between the distributions $p_y$ and $\hat{p}_{f,x}$, as measured by the cross-entropy function.

$$\ell(y, f(x)) = H(p_y, \hat{p}_{f,x}) \triangleq - \sum\nolimits_{i \in [L]} p_y(i) \cdot \log \hat{p}_{f,x}(i).$$

## 3.2 MODEL DISTILLATION

Distillation is a celebrated training techniques that utilizes one model's scores to train another model (Bucilă et al., 2006; Hinton et al., 2015). The former model is typically referred to as the 'teacher' model while the latter model is called the 'student' model. During distillation, given a teacher model $g : \mathcal{X} \to \mathbb{R}^L$ and an example $(x, y) \in \mathcal{X} \times \mathcal{Y}$, one first defines the teacher (softmax) distribution $\hat{p}_{g,x}(i)$ as per (2). Now, as opposed to utilizing the 'one-hot' label distribution $p_y$, we treat $\hat{p}_{g,x}(i)$ as the *pseudo* label distribution and define the *distillation version* of the softmax cross-entropy loss for the student model $f : \mathcal{X} \to \mathbb{R}^L$ as $\ell_{\text{distill}}(g(x), f(x)) = H(\hat{p}_{g,x}, \hat{p}_{f,x})$. For $a, b \in \mathbb{R}_+$, distillation involves minimizing

$$\frac{a}{n} \sum\nolimits_{i \in [n]} \ell(y_i, f(x_i)) + \frac{b}{n} \sum\nolimits_{i \in [n]} \ell_{\text{distill}}(g(x_i), f(x_i)) \tag{3}$$

Note that the objective in (3) utilized both the true labels $\{y_i\}$ and the teacher scores $\{g(x_i)\}$. When $b = 0$, this reduces to the standard training. More interestingly, when $a = 0$, (3) correspond to training with solely $\hat{p}_{g,x}(i)$ – a fairly common way to utilize distillation (Menon et al., 2020a).

**Remark 1.** For distillation, teacher models are usually much more complex and powerful as compared to student models (Bucilă et al., 2006; Hinton et al., 2015). However, the distillation with an equally complex, and even a simpler, teacher model has been shown to improve the quality of the student model via distillation (see, e.g., Rusu et al., 2016; Furlanello et al., 2018; Yuan et al., 2020).

## 4 DISTILLATION FOR TWO-STAGE INFERENCE

We now propose various distillation approaches to power the proposed two-stage inference framework. Recall that we intend to obtain a lightweight student model that can generate highly accurate predictions on easy instances and route hard instances to the large teacher model. This raises an interesting question if one needs to modify the distillation process in any way whatsoever to aid our objective as typically distillation envisions the student model to be used as a standalone model.

Towards this, we explore a general distillation framework that partitions the training examples $S$ into two groups: 1) easy instances $S_{\text{easy}}$ and 2) hard instances $S_{\text{hard}}$. Accordingly, we modify the distillation process such that the student incurs larger loss when it makes incorrect predictions $S_{\text{easy}}$. Furthermore, we penalize the student less for making mistakes on $S_{\text{hard}}$, which we accomplish by carefully designed supervision during the distillation. Now, we present two specific realizations of the above generic distillation framework based on two different strategies to partition the training examples into $S_{\text{easy}}$ and $S_{\text{hard}}$.

## 4.1 CLASS-SPECIFIC DISTILLATION

Many real-world data distributions exhibit a long-tail behavior where most of the instances we observe belong to a small number of classes, and the remaining classes are represented by very

---

[1]For brevity, we do not explicitly represent the temperature parameter in the rest of the paper.

few instances. This has sparked a long line of work on improving model performance on the tail classes (see, e.g., Cui et al., 2019; Cao et al., 2019; Kang et al., 2020; Menon et al., 2021). In contrast, we explore an orthogonal direction and leverage the data imbalance to enable efficient inference with large models. In particular, for $\mathcal{L}_{\text{in}} \subseteq [L]$, we define

$$S_{\text{easy}} = \{(x, y) \in S : y \in \mathcal{L}_{in}\} \quad \text{and} \quad S_{\text{hard}} = S \backslash S_{\text{easy}} = \{(x, y) \in S : y \notin \mathcal{L}_{in}\}.$$

Thus, we require the lite model to perform well only on a subset of classes $\mathcal{L}_{\text{in}}$ and route the examples from the remaining classes to the large model. Here, we hypothesize that the smaller model can better utilize its limited capacity to perform well on a subset of classes. Now, setting $\mathcal{L}_{\text{in}}$ to be the head classes will ensure that the lite model itself tries to predict the examples from the head classes. Since the underlying data-distribution is long-tail, only the examples from the tail classes (and a few hard examples from the head classes depending on the exact implementation details described later) are sent to the large model during inference.

With this general approach in mind, we propose a class-specific distillation approach.[2] Now, given a large teacher model $g$ and example $(x, y)$, we define a pseudo label distribution $\widetilde{p}_{g,x}^{\text{class}}$ as follows:

$$\widetilde{p}_{g,x}^{\text{class}} = \begin{cases} \hat{p}_{g,x} & \text{if } y \in \mathcal{L}_{\text{in}}, \\ (1 - \alpha) \cdot p_y + \frac{\alpha}{L} \cdot \mathbf{1} & \text{if } y \in [L] \backslash \mathcal{L}_{\text{in}}, \end{cases} \tag{4}$$

where $\hat{p}_{g,x}$ and $p_y$ denote the teacher's softmax distribution and the one-hot label distribution, respectively. In addition, $\alpha \in [0, 1]$ denotes a label-smoothing parameter. Now, we train a lite student $f : \mathcal{X} \to \mathbb{R}^L$ with the distillation loss

$$\ell_{\text{distill}}^{\text{class}}\big(g(x), f(x)\big) \triangleq H\big(\widetilde{p}_{g,x}^{\text{class}}, \hat{p}_{f,x}\big). \tag{5}$$

Note that the loss in (5) utilizes teacher softmax distribution for classes in $\mathcal{L}_{\text{in}}$ and relies on label-smoothed one-hot distribution for the remaining classes. This loss has two desirable properties for our two-stage inference objective: 1) As a result of standard distillation, the lite student behaves as a well-calibrated model with good performance on the examples from the classes in $\mathcal{L}_{\text{in}}$. 2) Due to standard label-smoothing (Szegedy et al., 2016; Müller et al., 2019), the lite student achieves a smaller margin on the examples belonging to the classes in $[L] \backslash \mathcal{L}_{\text{in}}$.

Let $f_{\mathcal{L}_{\text{in}}}$ be the lite student model obtained by minimizing the loss in (5). Now, given a test instance $x \in \mathcal{X}$, we first run inference with the lite model to obtain $f_{\mathcal{L}_{\text{in}}}(x)$. Subsequently, we decide if we need to make the final prediction based on $f_{\mathcal{L}_{\text{in}}}(x)$ or delegate the example to the large teacher $g$ to obtain the final prediction. We identify two useful delegation schemes, which we detail next.

**Class-based delegation.** Recall that the class-specific distillation aims to utilize the lite model to classify only the examples belonging to the classes in $\mathcal{L}_{in}$. Thus, the prediction made by $f_{\mathcal{L}_{\text{in}}}$ serves as a natural candidate for the delegation. In particular, when

$$\hat{y}_{\text{student}}(x) \triangleq \arg\max_{j \in [L]} f_{\mathcal{L}_{\text{in}}}(x) \in \mathcal{L}_{\text{in}},$$

we declare $\hat{y}_{\text{student}}(x)$ as the final prediction. Otherwise, $x$ is sent to the teacher and $\hat{y}_{\text{teacher}}(x) \triangleq \arg\max_{j \in [L]} g(x)$ becomes the final prediction.

**Margin-based delegation.** The class-based delegation is designed under the assumption that the lite model achieves high accuracy on the examples belonging to the classes in $\mathcal{L}_{\text{in}}$ and identifies the examples from $[L] \backslash \mathcal{L}_{\text{in}}$ with high fidelity. Both of these assumptions don't always hold in practice. In particular, $f_{\mathcal{L}_{\text{in}}}$ may find some instances from $\mathcal{L}_{\text{in}}$ hard and incorrectly predict a wrong class in $\mathcal{L}_{\text{in}}$ when presented with those instances. Similarly, $f_{\mathcal{L}_{\text{in}}}$ may predict a class from $\mathcal{L}_{\text{in}}$ when the test instance belongs to $[L] \backslash \mathcal{L}_{\text{in}}$.

Recall that the distillation loss in (5) is designed to ensure that $f_{\mathcal{L}_{\text{in}}}$ is well-calibrated on the instances from $\mathcal{L}_{\text{in}}$; as a result, it attains small margin on hard instances from $\mathcal{L}_{\text{in}}$. Moreover, by design, $f_{\mathcal{L}_{\text{in}}}$ realizes small margin on the instances from $[L] \backslash \mathcal{L}_{\text{in}}$. Thus, one can utilize the margin of the lite model $f_{\mathcal{L}_{\text{in}}}$ to perform delegation. Towards this, recall that

$$\gamma_{f_{\mathcal{L}_{\text{in}}}}(x) \triangleq \hat{p}_{f_{\mathcal{L}_{\text{in}}},x}([1]) - \hat{p}_{f_{\mathcal{L}_{\text{in}}},x}([2]) \tag{6}$$

---

[2]In what follows, without loss of generality, we assume $\mathcal{L}_{\text{in}} = [L']$, for $L' \leq L$. One can easily express our approach for general $\mathcal{L}_{\text{in}}$ with slightly cumbersome notation.

denotes the margin realized by $f_{\mathcal{L}_{\text{in}}}$ on $x$. Here, $\hat{p}_{f_{\mathcal{L}_{\text{in}}},x}([i])$ denotes the $i$-th largest element of the vector $\hat{p}_{f_{\mathcal{L}_{\text{in}}},x}$. Now, given a design parameter $\rho \in (0,1)$, we assign the following final prediction to a test instance $x \in \mathcal{X}$.

$$\hat{y}(x) = \begin{cases} \hat{y}_{\text{student}}(x) & \text{if } \gamma_{f_{\mathcal{L}_{\text{in}}}}(x) \geq \rho, \\ \hat{y}_{\text{teacher}}(x) & \text{if } \gamma_{f_{\mathcal{L}_{\text{in}}}}(x) < \rho. \end{cases} \tag{7}$$

The margin-based delegation aims to ensure that hard instances from $\mathcal{L}_{\text{in}}$ as well as all instances in $[L] \backslash \mathcal{L}_{\text{in}}$ get routed to the teacher for the final prediction.

**Remark 2.** The distillation based on the loss in (5) constitutes only one of many possible ways to transfer the performance of a teacher over a subset of classes to a lite student. We discuss two other class-specific distillation-based approaches in Sec. A of the appendix and evaluate those in Sec. 5.

### 4.2 MARGIN-BASED DISTILLATION

As discussed before, the margin assigned to an instance by a model is a natural proxy for the hardness of the instance, as viewed by the model. In Sec. 4.1, we utilize the margins assigned by the student to delegate the examples to the teacher. This raises an interesting question if one can utilize the margins to partition the training data into $S_{\text{easy}}$ and $S_{\text{hard}}$ example during the distillation. Towards this, given a teacher $g$ and a parameter $\rho_{\text{tr}} \in (0,1)$, we add a training example $(x,y)$ to $S_{\text{easy}}$ iff

$$\gamma_g(x) \triangleq \hat{p}_{g,x}([1]) - \hat{p}_{g,x}([2]) > \rho_{\text{tr}}. \tag{8}$$

Given this data partition, for an example $(x,y)$, we define the pseudo label distribution as follow:

$$\widetilde{p}_{g,x}^{\text{margin,L}} = \begin{cases} (\hat{p}_{g,x}(1), \ldots, \hat{p}_{g,x}(L)) & \text{if } (x,y) \in S_{\text{easy}} \\ (1-\alpha) \cdot p_y + \frac{\alpha}{L} \cdot \mathbf{1} & \text{otherwise .} \end{cases} \tag{9}$$

Now, we obtain a (lite) student $f_{\rho_{\text{tr}}}^L : \mathcal{X} \to \mathbb{R}^L$ by minimizing

$$\ell_{\text{distill}}^{\text{margin,L}}\big(g(x), f(x)\big) = H(\widetilde{p}_{g,x}^{\text{margin,L}}, \hat{p}_{f,x}). \tag{10}$$

For the two-stage inference, given a test instance $x$, we make the following final prediction.

$$\hat{y}(x) = \begin{cases} \hat{y}_{f_{\rho_{\text{tr}}}^L} = \arg\max_j f_{\rho_{\text{tr}}}^L(x)_j & \text{if } \gamma_{f_{\rho_{\text{tr}}}^L}(x) \geq \rho, \\ \hat{y}_{\text{teacher}}(x) & \text{otherwise,} \end{cases}$$

where $\hat{y}_{f_{\rho_{\text{tr}}}^L} = \arg\max_j f_{\rho_{\text{tr}}}^L(x)_j$.

**Remark 3.** For a small value of $\rho_{\text{tr}}$, we expect the student to make the correct prediction on almost all examples. Thus, the margin-based distillation proposed above becomes very similar to the normal distillation discussed in Sec. 3.2. On the other hand, for large values of $\rho_{\text{tr}}$, the student is expected to do well on only a small subset of examples during training and be able to identify the rest of the examples as hard instances that need to be routed to the teacher.

Similar to class-specific distillation, there are multiple potential variants of the margin-based distillation. We discuss one such variant that relies on an 'abstain' class in Sec. C of the appendix.

## 5 EXPERIMENTS

We now conduct a comprehensive empirical study of our distillation-based two-stage inference procedure. In particular, we evaluate various distillation frameworks introduced in Sec. 4 along with different choices of delegation methods. On standard image classification tasks, we establish that:

(i) A large improvement in accuracy over the student-only approach can be realized by delegating only a small fraction of examples to the large teacher model (Sec. 5.1). This validates our claim that one can rely on a much smaller student and achieve an accuracy similar to the teacher-only approach with a small increment in inference cost.

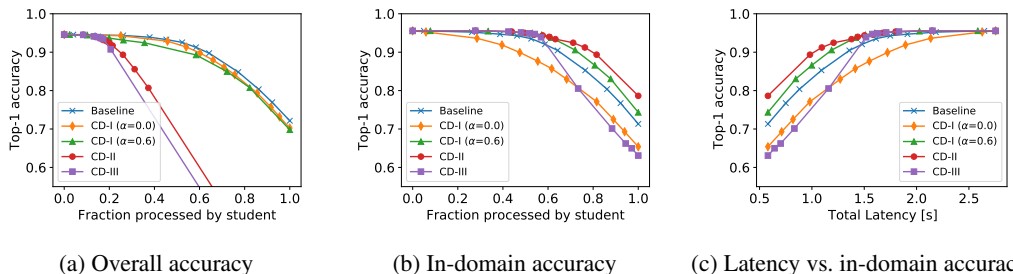

| (a) Overall accuracy | (b) In-domain accuracy | (c) Latency vs. in-domain accuracy |

Figure 2: Comparison of various *class-specific* distillation methods on CIFAR-100. Baseline denotes the standard distillation from (3). CD-I, CD-II, and CD-III denote the class-specific distillation approaches defined in Sec. 4.1, Sec. A.1, and Sec. A.2, respectively, with $|\mathcal{L}_{in}| = L' = 30$. Accordingly, we compute in-domain accuracy on test instances from the 30 classes in $\mathcal{L}_{in}$. Here, each lite student (ResNet-32) employs *margin-based* delegation to the teacher. The right-most plot depicts the (inference) latency vs. in-domain accuracy trade-off for the two-stage inference procedure.

(ii) Advantages of the two-stage inference are even more pronounced if we focus on the in-domain performance, where the in-domain portion of the instance space corresponds to a subset of classes or instances with a large margin (based on the large teacher model). This validates the utility of two-stage inference for those real-world settings where a large data imbalance exists in favor of in-domain instances (Sec. 5.2).

(iii) Class-specific distillation defined by (5) indeed achieves the desired behavior where the student enables a clear dichotomy among the in-domain and out-of-domain instances. By varying the label-smoothing parameter, we can improve in-domain model performance and delegate a small number of instances to the teacher at the cost of performance on the entire test data (Sec. 5.2).

Furthermore, by delegating a small fraction of hard instances to a large teacher model, our proposed class-specific distillation and a variant of margin-based distillation enable efficient inference on sentence classification and reading comprehension tasks in the NLP domain, respectively (Sec. 5.3).

We mainly focus on three benchmark image datasets – CIFAR-100 (Krizhevsky, 2009), ImageNet-1k (Russakovsky et al., 2015), and ImageNet-21k (Deng et al., 2009). In addition, we also evaluate the proposed two-stage inference procedure on a sentence classification task based on MNLI (Williams et al., 2018) and a reading comprehension task based on SQuAD dataset (Rajpurkar et al., 2016). On image classification tasks, we use EfficientNet-L2 (Xie et al., 2020) as the large teacher. As for the lite student, we utilize ResNet (He et al., 2016a;b) and MobileNetV3 (Howard et al., 2019) for CIFAR-100 and ImageNet, respectively. For the sentence classification and reading comprehension tasks, RoBERTa-Large (Liu et al., 2019) and T5-11B (Raffel et al., 2019) serve as the teachers, respectively. We use MobileBERT (Sun et al., 2020) as the lite student for both MNLI and SQuAD. We provide a detailed description in Sec. D of the appendix.

## 5.1 OVERALL ACCURACY GAINS

We begin by establishing the utility of the two-stage inference procedure for overall performance improvement. Towards this, Fig. 2a, 3a, and 5a (in appendix) depict the two-stage overall accuracy achieved by various class-specific distillation approaches via *margin-based* delegation. Besides, we also include conventional model distillation as *Baseline* in our evaluation. Note that, for all approaches, allowing the lite student model to delegate hard instances (the ones with a small margin) to the large teacher significantly increases the accuracy as compared to the performance of (one-stage) student-only inference approach. Focusing on CIFAR-100, both Baseline and CD-I can approximate the performance of the teacher model by delegating only ∼40% test instance, which translates to ∼60% reduction in the inference cost as compare to the large teacher (cf. Fig. 2a). Here, we note that CD-II and CD-III train the student to make the correct prediction on only the instances belonging to the classes in $\mathcal{L}_{in}$, leading to poor overall accuracy on the entire test set. However, unsurprisingly, as student delegates more instances to the teacher, the two-stage overall accuracy improves. Similar conclusions also hold on ImageNet datasets (cf. Fig. 3a and 5a). Table 1 and 4 (in the appendix) show that the class-specific distillation based two-stage inference leads to similar improvements in the overall accuracy when we employ *class-based* delegation from Sec. 4.1.

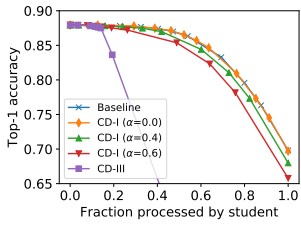 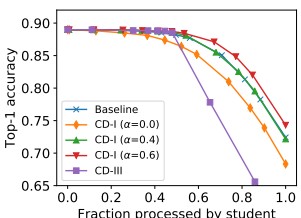 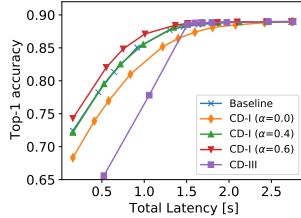

(a) Overall accuracy       (b) In-domain accuracy       (c) Latency vs. in-domain accuracy

Figure 3: Comparison of various *class-specific* distillation methods on ImageNet-1k. Baseline denotes the standard distillation from (3). CD-I and CD-III denote the class-specific distillation approaches defined in Sec. 4.1 and Sec. A.2, respectively, with $|\mathcal{L}_{\text{in}}| = L' = 300$. Accordingly, we compute in-domain accuracy on the test instances from the 300 classes in $\mathcal{L}_{\text{in}}$. Here, each lite student (MobileNetV3-0.75) employs *margin-based* delegation. The right-most plot depicts the (inference) latency vs. in-domain accuracy trade-off for the two-stage inference procedure.

## 5.2 IN-DOMAIN PERFORMANCE

Next, we verify that two-stage distillation is even more beneficial in those real-world settings where ML models encounter heavily imbalanced data during the inference. Ideally, the lite student model should make a prediction on frequent but easy instances and delegate rare but hard instances to the large teacher. This would ensure that the two-stage inference procedure realizes high overall accuracy (compared to inference with only the student) while significantly lowering the inference cost (compared to inference with only the teacher). With this in mind, we evaluate the performance of the two-stage inference procedure on in-domain (so-called easy instances). By design, for class-specific distillation, in-domain instances belong to the classes in $\mathcal{L}_{\text{in}}$. Similarly, for the margin-based distillation, in-domain instances are the ones where teacher assigns a large margin $\rho$.

| | Approach | In-domain | | Overall | |
|---|---|---|---|---|---|
| | | Accuracy | Fraction | Accuracy | Fraction |
| ResNet-32 | Baseline | 0.71 | 1.00 | 0.72 | 1.00 |
| | CD-I ($\alpha = 0.0$) | 0.88 | 0.74 | 0.88 | 0.26 |
| | CD-I ($\alpha = 0.6$) | 0.88 | 0.84 | 0.86 | 0.32 |
| | CD-II | 0.78 | 1.00 | 0.24 | 1.00 |
| | CD-III | 0.91 | 0.69 | 0.90 | 0.25 |
| ResNet-56 | Baseline | 0.75 | 1.00 | 0.75 | 1.00 |
| | CD-I ($\alpha = 0.0$) | 0.89 | 0.77 | 0.90 | 0.27 |
| | CD-I ($\alpha = 0.6$) | 0.90 | 0.82 | 0.88 | 0.29 |
| | CD-II | 0.80 | 1.00 | 0.24 | 1.00 |
| | CD-III | 0.92 | 0.71 | 0.90 | 0.25 |

Table 1: Performance of two-stage inference procedure on CIFAR-100. The student employs class-specific distillation with $|\mathcal{L}_{\text{in}}| = L' = 30$, with in-domain referring to the instances from the classes in $\mathcal{L}_{\text{in}}$. During inference we, use an appropriate *class-based* delegation method. See Fig. 2 for the identity of the distillation approaches. Fraction denotes the fraction of test instances where the student model makes the final prediction. Unlike margin-based delegation, for given teacher and student models, we obtain a single value of (Accuracy, Fraction) tuple with class-based delegation.

the classes in $\mathcal{L}_{\text{in}}$. Similarly, for the margin-based distillation, in-domain instances are the ones where teacher assigns a large margin $\rho$.

For CIFAR-100, Fig. 2b shows that two-stage inference achieves much better in-domain (defined by $|\mathcal{L}_{\text{in}}| = 30$ classes) accuracy as compared to overall accuracy. This implies that in a real-world setting where in-domain instances are frequent, the two-stage inference can efficiently achieve very good overall performance by having the student predict most of the in-domain instances. Note that overall performances in such a scenario (with data-imbalance) is not captured by Fig. 2a which is based on fair balanced test data. Instead of aiming to imitate the data imbalance encountered in practice, we have separately highlighted the performance of two-stage inference on the in-domain instances. This shows that the more imbalanced the data is the more advantageous two-stage distillation would be in terms of saving the inference cost without affecting the classification performance. As per Fig 2c, we essentially maintain same accuracy as the teacher but reduce latency by more than 2x. Here, we also note that Fig. 2a implies that two-stage inference would indeed enable one to approximate the teacher performance on all instances, whether they belong to in-domain or not.

Interestingly, Fig. 2 indicates that CD-I with margin-based delegation gives a favorable tradeoff between in-domain and (balanced) overall accuracy. The label-smoothing parameter $\alpha$ in (4) helps create a *dichotomy* between in-domain and out-of-domain instances. The larger values of $\alpha$ ensure that the student assigns a smaller margin to out-of-domain examples and tries to improve its performance on in-domain instances. This is reflected in Fig. 2b where CD-I with large $\alpha$ achieves higher

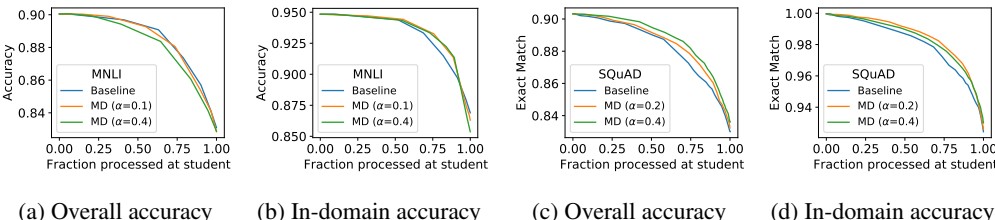

| (a) Overall accuracy | (b) In-domain accuracy | (c) Overall accuracy | (d) In-domain accuracy |

Figure 4: Performance of the proposed two-stage inference on NLP tasks. **Left half:** Overall and in-domain accuracy of the margin-based distillation coupled with margin-based delegation on MNLI. Here baseline corresponds to the two-stage inference enabled by the standard distillation (cf. (3)). **Right half:** Overall and in-domain accuracy (exact match) comparison of margin-based distillation on SQuAD task. Here, Baseline denotes a normally trained student used in combination with the teacher. Note that MD denotes a margin-based distillation with label smoothing.

accuracy by predicting a larger fraction of in-domain instances at the student. We note the similar trend in Table 1 where we combine class-specific distillation with class-based delegation

We also evaluate the in-domain two-stage performance of class-based distillation on ImageNet-1k with $|\mathcal{L}_{\text{in}}| = 300$. The conclusions from Fig. 3b, 3c and Table 4 (in the appendix) are similar to those observed on CIFAR-100. Also, see Fig. 5b (in the appendix) for the identical trends on ImageNet-21k. Finally, we also studied the in-domain performance of two-stage inference enabled by margin-based distillation from Sec. 4.2 on both CIFAR-100 and ImageNet-1k in the appendix.

### 5.3 Two-stage inference in NLP domain

**Sentence classification task.** Fig. 4a and 4b show the performance of our proposed two-stage inference framework on MNLI. Since it has only 3 classes, we employ margin-based distillation (with $\rho_{\text{tr}} = 8.0$) from (9) with margin-based delegation. Our conclusion for the image classification also extends to the text domain and the two-stage inference framework enables a lite student to leverage high-quality teacher by delegating a small portion of (hard) instances to the teacher. We provide inference-latency vs. in-domain performance trade-off for MNLI in the appendix (cf. Fig. 8a).

**Reading comprehension task.** To showcase the generality of our approach , we apply the two-stage inference method to a machine comprehension task (SQuAD), where there is a fundamental mismatch between student architecture which is based on span selection whereas teacher is based on encoder-decoder architecture. Thus, one cannot do standard distillation by transferring logits from teacher to students, but a suitable modification of margin-based distillation from (9) still works. We partition the training examples into easy and hard instances by thresholding teacher's log-likelihood of ground-truth answer given the context. While training student, for easier examples, we use one hot labels for the start and end span as no teacher logits are available. Whereas for hard examples, we still employ label smoothing. We try two values $\alpha = 0.2, 0.4$ in the experiments. Our results in Fig. 4d and 4d, which has similar conclusions to the classification experiments. See Fig. 8b for the inference-latency vs. in-domain performance trade-off achieved by the two-stage inference.

## 6 Discussion

We propose a *distillation-based* two-stage inference framework to efficiently leverage large models with prohibitively large inference costs in real-world settings. Given a large model, we distill a lite student that utilizes its limited model capacity to perform well on easy (in-domain) instances and can identify hard (out-of-domain) instances. When deployed in tandem, the lite model generates the final prediction on easy but frequent instances and delegates hard but rare instances to the large model. This ensures much higher performance (compared to inference based on only the lite model) and a much smaller inference cost (compared to inference with only the large model). We propose various distillation methods to enable such two-stage inference. We establish the utility of these approaches for realizing efficient and accurate inference on both image classification and NLP benchmarks.

Generalizing our framework to design a multi-stage inference procedure is a natural direction for future research. To enhance the applicability of our method to wider settings, another research direction would be to devise a principled approach for applying our method to other architectures including non-parametric models like k-NN, generalizing what we did for reading comprehension.

REPRODUCIBILITY STATEMENT

In Sec. 5 and Sec. D (in the appendix), we provide detailed descriptions of the experimental setup used in this paper, including the pre-trained (teacher) models, student models, datasets, and how to train and evaluate our two-stage framework. Also, since we are only training small student models via standard and proposed distillation methods, it is relatively cheap to reproduce our experimental results.

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

# A  VARIANTS OF CLASS-SPECIFIC DISTILLATION

Here, we present two additional variants of class-specific distillation that enable transferring the performance of a large teacher model to a lite student model on a subset of class. Subsequently, the lite student can be employed in our two-stage inference framework.

## A.1  IN-DOMAIN CLASS-DISTILLATION

Since we do not intend the student to make the final prediction on an instance from $[L]\backslash\mathcal{L}_{\text{in}}$, we can train the student to perform an $|\mathcal{L}_{\text{in}}| = |L'|$-way classification. In particular, given a teacher $g$ and example $(x, y)$, we define a pseudo-label distribution:

$$\widetilde{p}_{g,x}^{\text{class},L'} = \begin{cases} \hat{p}_{g,x}^{L'} \in [0,1]^{L'} & \text{if } y \in \mathcal{L}_{\text{in}}, \\ \frac{1}{L'} \cdot \mathbf{1} \in \mathbb{R}^{L'} & \text{if } y \in [L]\backslash\mathcal{L}_{\text{in}}. \end{cases} \tag{11}$$

Here, $\hat{p}_{g,x}^{L'}$ denotes the softmax distribution restricted to $\mathcal{L}_{\text{in}}$:

$$\hat{p}_{g,x}^{L'}(j) = e^{g(x)_j} / \sum\nolimits_{i \in \mathcal{L}_{\text{in}}} e^{g(x)_i} \text{ for } j \in \mathcal{L}_{\text{in}}. \tag{12}$$

Now we get the student $f_{\mathcal{L}_{\text{in}},L'} : \mathcal{X} \to \mathbb{R}^{L'}$ by minimizing

$$\ell_{\text{distill}}^{\text{class},L'}\big(g(x), f(x)\big) = H\big(\widetilde{p}_{g,x}^{\text{class},L'}, \hat{p}_{f,x}\big) \tag{13}$$

A lite model $f_{\mathcal{L}_{\text{in}},L'}$ trained with the loss in (13) aims to classify an instance from $\mathcal{L}_{\text{in}}$ to the correct class. At the same time, such a model is also trained to generate a non-informative uniform distribution $\frac{1}{L'} \cdot \mathbf{1}$ on the instances from $[L]\backslash\mathcal{L}_{\text{in}}$, which by definition corresponds to zero margin.

Now, given the student $f_{\mathcal{L}_{\text{in}},L'}$ and teacher $g$, one can define a two-stage inference by using margin-based delegation. In particular, for a test instance $x \in \mathcal{X}$, the final prediction becomes

$$\hat{y}(x) = \begin{cases} \arg\max_{j \in [L']} f_{\mathcal{L}_{\text{in}},L'}(x)_j & \text{if } \gamma_{f_{\mathcal{L}_{\text{in}},L'}}(x) \geq \rho, \\ \arg\max_{j \in [L]} g(x)_j & \text{if } \gamma_{f_{\mathcal{L}_{\text{in}},L'}}(x) < \rho, \end{cases}$$

where $\gamma_{f_{\mathcal{L}_{\text{in}},L'}}(x)$ is the margin assigned to $x$ by the student $f_{\mathcal{L}_{\text{in}},L'}$.

## A.2  IN-DOMAIN CLASS-DISTILLATION WITH 'ABSTAIN' OPTION.

We next explore another natural candidate for class-specific distillation, where the lite student aims to correctly classify an instance $x$ from $\mathcal{L}_{\text{in}}$ and declares to 'abstain' on the instances from $[L]\backslash\mathcal{L}_{\text{i}}$. To realize this, we train the student to perform an $(L'+1)$-way classification: Given a teacher $g$ and example $(x, y)$, we define a pseudo label distribution:

$$\widetilde{p}_{g,x}^{\text{class},L'+1} = \begin{cases} (\hat{p}_{g,x}^{L'}, 0) \in [0,1]^{L'+1} & \text{if } y \in \mathcal{L}_{\text{in}}, \\ (0, \ldots, 0, 1) & \text{if } y \in [L]\backslash\mathcal{L}_{\text{in}}, \end{cases}$$

where $\hat{p}_{g,x}^{L'}$ denotes the softmax distribution restricted to $L'$ classes, as defined in (12). Now, one can perform the distillation by utilizing $\widetilde{p}_{g,x}^{\text{class},L'+1}$, i.e., minimize

$$\ell_{\text{distill}}^{\text{class},L'+1}\big(g(x), f(x)\big) = H\big(\widetilde{p}_{g,x}^{\text{class},L'+1}, \hat{p}_{f,x}\big). \tag{14}$$

Note that (14) encourages the trained lite model $f_{\mathcal{L}_{\text{in}},L'+1}$ to predict $(L'+1)$-th class, i.e., 'abstain' class, for all instances from $[L]\backslash\mathcal{L}_{\text{in}}$. This leads to a natural *abstain class-based delegation* approach for two-stage inference. Let

$$\hat{y}_{\text{student}}^{\text{abstain}}(x) = \arg\max_{j \in [L'+1]} f_{\mathcal{L}_{\text{in}},L'+1}(x)_j.$$

Then, given $f_{\mathcal{L}_{\text{in}},L'+1}$ and $g$, one makes the following final prediction for a test instance.

$$\hat{y}(x) = \begin{cases} \hat{y}_{\text{student}}^{\text{abstain}}(x) & \text{if } \hat{y}_{\text{student}}^{\text{abstain}}(x) \leq L', \\ \hat{y}_{\text{teacher}}(x) & \text{if } \hat{y}_{\text{student}}^{\text{abstain}}(x) = L' + 1. \end{cases}$$

In addition, analogous to (7), one can also define a two-stage inference procedure via margin-based delegation.

**Remark 4.** Note that using an 'abstain' class is closely related to classification with a reject option (see Sec. B). However, as opposed to the traditional classification with the reject paradigm, we intend to provide supervision for the so call reject class as well.

## B    CLASSIFICATION WITH A REJECT OPTION

There is a large literature on selective classification, also known as classification with a reject option or abstention Grandvalet et al. (2009); Bartlett & Wegkamp (2008); Cortes et al. (2016); Geifman & El-Yaniv (2017); Ramaswamy et al. (2018); Ni et al. (2019). Here, one seeks a predictor $f \colon \mathcal{X} \to \mathcal{Y} \cup \{\perp\}$, where a prediction of $\perp$ denotes the classifier is uncertain. To avoid the degenerate solution of abstaining on all samples, one assumes a fixed rejection cost $c \in (0, 1]$. The goal is to then trade-off the misclassification error on non-abstained samples with the total cost incurred on abstained samples, i.e., minimize the loss

$$\ell(y, f(x)) = \mathbb{1}_{y \neq f(x) \wedge f(x) \neq \perp} + c \cdot \mathbb{1}_{f(x) = \perp}. \tag{15}$$

The Bayes-optimal classifier for this objective abstains on samples with high uncertainty on the "true" label, i.e., Ramaswamy et al. (2018)

$$f^*(x) = \begin{cases} \perp & \text{if } \max_{y \in \mathcal{Y}} \mathbb{P}(y \mid x) \leq 1 - c \\ \underset{y \in \mathcal{Y}}{\operatorname{argmax}} \, \mathbb{P}(y \mid x) & \text{else.} \end{cases} \tag{16}$$

## C    VARIANT OF MARGIN-BASED DISTILLATION: USING ABSTAIN CLASS

Another natural approach for margin-based distillation is to utilize an 'abstain' class to encourage student to not spend its model capacity on correctly classifying hard instances (where teacher achieves a low-margin). Towards this, we can define the following pseudo label distribution for an example $(x, y)$.

$$\widetilde{p}_{g,x}^{\text{margin}} = \begin{cases} (\hat{p}_{g,x}(1), \ldots, \hat{p}_{g,x}(L), 0) & \text{if } (x, y) \in S_{\text{easy}} \\ (0, \ldots, 0, 1) \in \{0, 1\}^{L+1} & \text{otherwise}. \end{cases}$$

Now, we distill a lite student model $f_{\rho_{\text{tr}}}^{L+1} : \mathcal{X} \to \mathbb{R}^{L+1}$ based on $\widetilde{p}_{g,x}^{\text{margin}}$, i.e., we minimize

$$\ell_{\text{distill}}^{\text{margin}}\big(g(x), f(x)\big) = H(\widetilde{p}_{g,x}^{\text{margin}}, \hat{p}_{f,x}) \tag{17}$$

Note that we train the student to perform an $(L + 1)$-way classification, where it aims to classify an example in $S_{\text{easy}}$ to one of $L$ original classes and the examples in $S_{\text{hard}}$ to the 'abstain' class. Now, one may employ $g$ and $f_{\rho_{\text{tr}}}^{L+1}$ to enable a two-stage inference procedure that makes the final prediction for a test instance $x \in \mathcal{X}$ as

$$\hat{y}(x) = \begin{cases} \hat{y}_{f_{\rho_{\text{tr}}}^{L+1}} & \text{if } \hat{y}_{f_{\rho_{\text{tr}}}^{L+1}} \leq L' \ \& \ \gamma_{f_{\rho_{\text{tr}}}^{L+1}}(x) \geq \rho, \\ \hat{y}_{\text{teacher}}(x) & \text{otherwise,} \end{cases}$$

where $\hat{y}_{f_{\rho_{\text{tr}}}^{L+1}} = \arg\max_j f_{\rho_{\text{tr}}}^{L+1}(x)_j$.

## D    DETAILS OF EXPERIMENTAL SETUP

**Datasets.**    We use three benchmark image datasets – CIFAR-100 (Krizhevsky, 2009), ImageNet ILSVRC 2012 (a.k.a. ImageNet-1k) (Russakovsky et al., 2015), and ImageNet-21k (Deng et al., 2009). CIFAR-100 contains 60k (50k train/10k test) images annotated with one of 100 object categories distributed uniformly. ImageNet (ILSVRC 2012), on the other hand, is a much larger dataset with 1.33M (1.28M train/50k test) images annotated with one of 1000 object categories also distributed uniformly. As for ImagenNet-21k, it originally contains images 12.8M from 21,843 classes. We select 17,203 classes with at least 100 images and create a balanced test set with 50 images from each of the selected classes. This remaining images from the selected classes provides us with a training set containing ∼11.8M images.

In addition to image datasets, we also evaluate the proposed distillation-based two-stage inference procedure on MNLI (Williams et al., 2018) and SQuAD datasets (Rajpurkar et al., 2016), which are

| Model | Parameters | FLOPs |
|-------|-----------|-------|
| ResNet-8 | 0.09 M | 16 M |
| ResNet-14 | 0.19 M | 30 M |
| ResNet-20 | 0.27 M | 44 M |
| ResNet-32 | 0.46 M | 72 M |
| ResNet-44 | 0.66 M | 100 M |
| ResNet-56 | 0.85 M | 128 M |

Table 2: (Lite) ResNet models for CIFAR-100.

| Model | Parameters | FLOPs |
|-------|-----------|-------|
| MobileNetv3-0.35 | 2.14 M | 40 M |
| MobileNetv3-0.50 | 2.70 M | 70 M |
| MobileNetv3-0.75 | 4.01 M | 156 M |
| MobileNetv3-1.00 | 5.50 M | 218 M |
| MobileNetv3-1.25 | 8.29 M | 358 M |

Table 3: (Lite) MobileNetV3 models for ImageNet.

standard sentence classification and QA benchmarks, respectively. MNLI dataset corresponds to a 3-way classification task with 392,702 training examples and a matched test set consisting of 9,815 instances. As for the SQuAD dataset, it is a span selection task over a sequence length of 384 with 87,599 and 10,570 train and test examples, respectively.

**Large teacher models.** For the image classification tasks, we use EfficientNet-L2 (Xie et al., 2020; Tan & Le, 2019; Foret et al., 2021) which is the state-of-art for image classification on multiple datasets. It is an optimized convolutional neural network pretrained on both ImageNet and unlabeled JFT-300M (Sun et al., 2017) with input resolution of 475. EfficientNet-L2 is a large model with 480M parameters which requires 478G FLOPs per inference and achieves an accuracy of 88.6% on ImageNet. For CIFAR-100, we used base EfficientNet-L2 model with a new fine-tuned classification layer achieving an accuracy of 94.4%. For the sentence classification task, a RoBERTa-Large (Liu et al., 2019) model serves as a teacher, which is pre-trained on a general purpose text corpus of 160 GB size. It is a transformer-based encoder model with 355M parameters that requires 155G FLOPs per inference while achieving an accuracy of 90.2% on MNLI.For the QA task requiring reading comprehension, we use T5-11B (Raffel et al., 2019) which exhibits competitive performance on a wide variety of NLP tasks. It is a text-to-text transformer architecture pretrained on a subset of the common crawl with sequence lengths of 512. T5-11B is a large model with 11B parameters and requires 6.6T FLOPs per inference achieving an accuracy of 90.2% on SQuAD.

**Lite student models.** For CIFAR-100, we use small ResNet networks (He et al., 2016a;b) whose details are listed in Table 2. As for ImageNet, we explore MobileNetV3 (Howard et al., 2019) of varying sizes as lite student models (see Table 3). For NLP tasks, we use MobileBERT (Sun et al., 2020) as the lite student model which has 25M parameters and requires 13.5G FLOPs per inference.

**Details of the experiment in Figure 1.** For this experiment CIFAR-100 dataset was used. To sweep full spectrum all the way till teacher size, we used ResNet networks as detailed in Table 2 as the student networks and the largest ResNet-56 as the teacher network. We used class-specific distillation from Sec 4.1.

## E  ADDITIONAL EXPERIMENTAL RESULTS

**Class-specific distillation with class-based delegation.** Table 4 represents the two-stage performance (both in-domain and overall) realized by class-specific distillation approaches (cf. Sec. 4.1) on ImageNet when we employ appropriate class-based delegation.

**Class-specific distillation on ImageNet-21k dataset.** Figure 5 shows the performance of our proposed distillation-based two-stage inference framework on ImageNet-21k dataset. As discussed in Sec. D, we work with 17,203 out of 21,843 classes originally present in the dataset. Since we don't have access to a high-performing teacher model on this dataset, we work with the *oracle* teacher (the one that knows the true label) to simulate the two-stage stage inference framework. Also, while training a MobileNetV3-0.75 model as a student, we use the one-hot labels as the supervision for the classes in $\mathcal{L}_{\text{in}}$ and utilize label-smoothing for the instances belonging to the remaining classes. Accordingly, Baseline here corresponds to a normally trained student (MobileNetV3-0.75 model) combined with the oracle teacher.

**Margin-based distillation with margin-based delegation.** Here, we study both overall and in-domain performance of two-stage inference enabled by margin-based distillation from Sec. 4.2

| | Approach | In-domain | | Overall | |
|---|---|---|---|---|---|
| | | Accuracy | Fraction | Accuracy | Fraction |
| MobileNet-0.75 | Baseline | 0.75 | 1.00 | 0.68 | 1.00 |
| | CD-I ($\alpha = 0.0$) | 0.85 | 0.76 | 0.84 | 0.23 |
| | CD-I ($\alpha = 0.4$) | 0.84 | 0.82 | 0.81 | 0.32 |
| | CD-I ($\alpha = 0.6$) | 0.83 | 0.86 | 0.78 | 0.37 |
| | CD-III | 0.87 | 0.63 | 0.85 | 0.23 |
| MobileNet-1.25 | Baseline | 0.79 | 1.00 | 0.72 | 1.00 |
| | CD-I ($\alpha = 0.0$) | 0.86 | 0.79 | 0.85 | 0.28 |
| | CD-I ($\alpha = 0.4$) | 0.85 | 0.83 | 0.83 | 0.31 |
| | CD-I ($\alpha = 0.6$) | 0.85 | 0.86 | 0.81 | 0.36 |
| | CD-III | 0.87 | 0.70 | 0.85 | 0.25 |

Table 4: Performance of two-stage inference procedure on ImageNet-1k. The student model is obtained by a class-specific distillation approach with $|\mathcal{L}_{\mathrm{in}}| = L' = 300$, with in-domain referring to the instances belonging to the classes in $\mathcal{L}_{\mathrm{in}}$. CD-I and CD-III denote the class-based distillation approaches defined in Sec. 4.1 and Sec. A.2. Baseline refers to the standard distillation from (3) with $a = 0$ and $b = 1$. The inference procedure employs an appropriate *class-based delegation* for each distillation approach. Fraction denotes the fraction of test instances where the student model makes the final prediction. Note that, for standard distillation (Baseline), the student cannot delegate any examples to the teacher via class-based delegation.

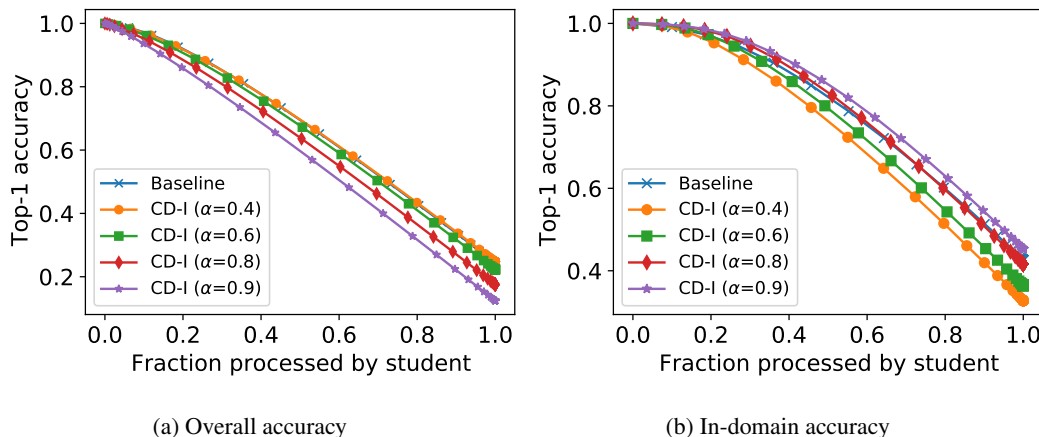

(a) Overall accuracy           (b) In-domain accuracy

Figure 5: Comparison of various class-specific distillation methods on (subset of) ImageNet-21k. As described in Sec. D, we work with a subset corresponding to 17,203 out of 21,843 classes. Baseline denotes the standard distillation from (3) with $a = 0$ and $b = 1$. CD-I denote the class-specific distillation approaches defined in Sec. 4.1. with $|\mathcal{L}_{\mathrm{in}}| = L' = 1000$. Accordingly, in-domain accuracy is computed via focusing on the test instances from the 1000 classes in $\mathcal{L}_{\mathrm{in}}$. Here, each lite student model (a MobileNetV3-0.75 model) employs margin-based delegation.

on both CIFAR-100 (cf. Fig. 6) and ImageNet (cf. Fig. 7). Here, in-domain instances correspond to those instances where the student assigns a margin of at least $\rho = 0.4$. As evident, margin-based distillation also leads to improved in-domain two-stage accuracy. However, Baseline and MD from (17) have very similar performance on both datasets.

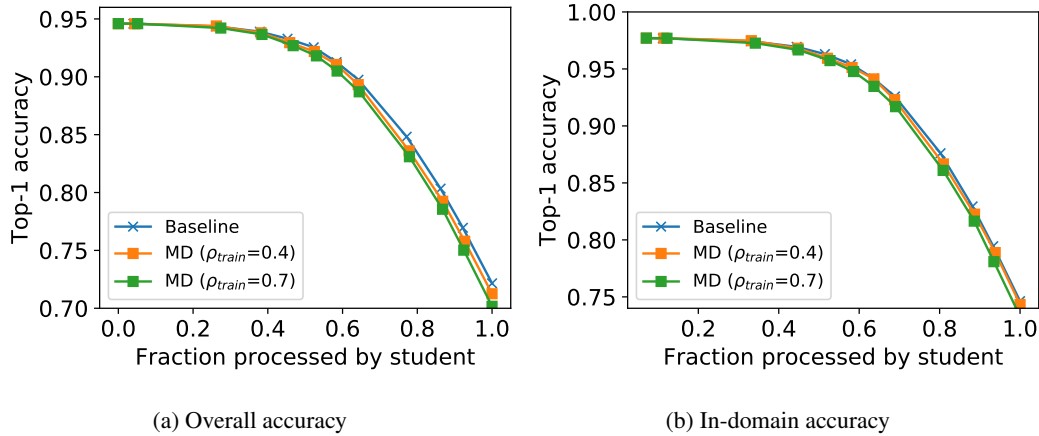

(a) Overall accuracy          (b) In-domain accuracy

Figure 6: Performance of two-stage inference procedure enabled by the margin-based distillation approach from Sec. 4.2 on CIFAR-100. Again, Baseline denotes the standard distillation from (3) with $a = 0$ and $b = 1$. MD represent the margin-based distillation approach from (17). Here, each lite student model (a ResNet-32 model) employs *margin-based delegation*. In-domain accuracy is computed on the test instances where student achieves a margin $\rho$ of at least 0.4.

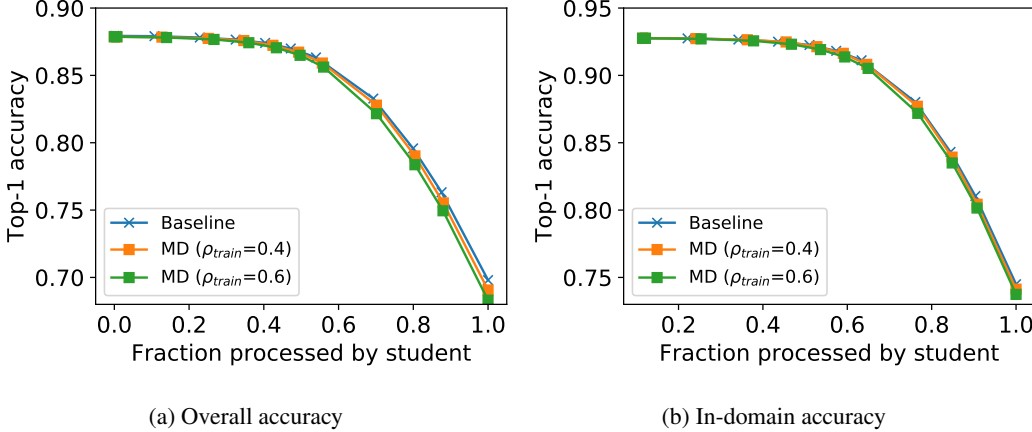

(a) Overall accuracy          (b) In-domain accuracy

Figure 7: Performance of two-stage inference procedure enabled by the margin-based distillation approach from Sec. 4.2 on ImageNet. Here, each lite student model (a MobileNetV3-0.75 model) employs *margin-based delegation*. In-domain accuracy is computed on the test instances where student achieves a margin $\rho$ of at least 0.4.

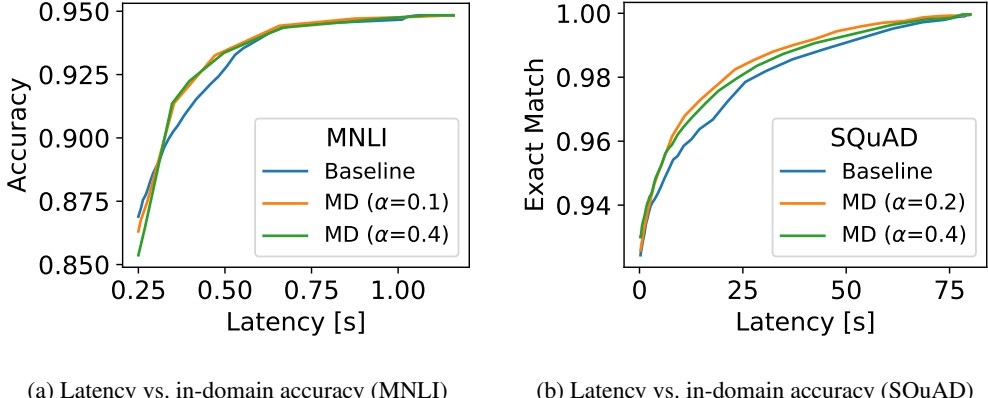

(a) Latency vs. in-domain accuracy (MNLI)       (b) Latency vs. in-domain accuracy (SQuAD)

Figure 8: The inference-latency vs. in-domain performance trade-off realized by the two-stage inference procedure on the NLP tasks. MD denotes the margin-based distillation with label smoothing parameter $\alpha$. **Left:** On MNLI dataset, RoBERTa-Large and MobileBERT are used as the teacher and student models, respectively. Here, Baseline corresponds to the two-stage inference enabled by the standard distillation (cf. (3)). **Right:** On SQuAD dataset, T5 and MobileBERT are used as the teacher and student models, respectively. Here, Baseline denotes a normally trained MobileBERT used in combination with T5 model.

