# OpenReview forum: "When in Doubt, Summon the Titans: A Framework for Efficient Inference with Large Models"
_ICLR.cc/2022/Conference — ICLR 2022 Submitted_

### Official Review · Reviewer_npvb · 2021-10-30

**Correctness:** 2
**Technical Novelty And Significance:** 1
**Empirical Novelty And Significance:** 2
**Recommendation:** 3
**Confidence:** 4

**Main Review:**

Strength: The paper is fairy well written and easy to follow.

Weaknesses:
There are multiple weaknesses. The idea of combining a small student model and the original large model for inference is pretty intuitive (I am not saying intuitive is a bad thing) but does not demonstrate well. It is not clear why such a method can be interesting and be better than existing methods. For example, Section 2 discusses some related work on efficient inference. However, these are never compare with the proposal method, neither theoretically nor empirically. Especially, in the experiments, only the standard distillation baseline is compared with the proposed method, which is quite limited and fail to demonstrate the advantages of the proposed method. Thus, the motivation of the proposed method is quite unclear. In addition, since the focus of this paper is on inference efficiency, it is necessary to compare the inference time, in addition to the current accuracy.

**Summary Of The Paper:**

This paper studies efficient inference problem for large models. It proposes to train a small student model, and performs inference for easy data on the student model, and for hard data on the original large model. Experiments show the proposed method performs better than the simple baseline of standard distillation model.

**Summary Of The Review:**

This paper fails to demonstrate the motivation and advantage of the proposed method for efficient inference of large model. It does not meet the acceptance bar of the conference.

---

> ### Author Response · Authors · 2021-11-19
> **Response to Reviewer npvb**
>
> Thank you for your effort in reviewing our submission. We are glad that the reviewer found our paper fairly well written and easy to follow.
>
> __The idea of combining a small student model and the original large model for inference is pretty intuitive…but does not demonstrate well.__
>
> We do agree that our proposed distillation-based two-stage inference framework is simple and intuitive. However, to the best of our knowledge, it has not been previously considered in the literature with or without *specialized* distillation. In our submission, we carry out an extensive evaluation of the proposed distillation-based two-stage inference framework on both image classification and NLP benchmarks. Our results indeed demonstrate that the simple and intuitive idea is highly effective in realizing favorable performance vs. inference cost trade-off (see, e.g., Figure 2, 3, and 4).
>
> __It is not clear why such a method can be interesting and be better than existing methods…__
>
> Regarding existing methods for efficient inference, as already mentioned in Section 2, Quantization and pruning techniques are complementary to our distillation-based exploration. Such techniques can naturally be utilized in our framework to further reduce the inference cost (e.g., by replacing teacher with its quantized/pruned version which is still larger than the lightweight student model).
>
> On the other hand, adaptive computation based approaches (e.g., early-exit networks) are highly specific to underlying task and model architecture. These approaches require designing an end-to-end training pipeline from scratch. In contrast, our modular approach aims to leverage existing (SoTA) large models (some of which might be only accessible by API) by only training cheap student models. Furthermore, the distillation based approach is not tied to any specific architecture. In fact, in our experiments on SQuAD dataset, we leverage an encoder-decoder teacher along with a encoder-only student.
>
> We believe that these aforementioned reasons show why our simple yet effective distillation-based two-stage inference framework is interesting in its own right.
>
> __Especially, in the experiments, only the standard distillation baseline is compared with the proposed method, which is quite limited and fail to demonstrate the advantages of the proposed method.__
>
> Note that, given a high-performing expensive large model for the underlying task, our objective is to approximate the performance of the large model without incurring high inference cost. Towards this, the two-stage inference framework (with standard distillation or with specialized distillation) can successfully realize a desirable performance vs. inference cost trade-off. (To the best of our knowledge, we are the first ones to consider distillation-based two-stage inference even with the standard distillation.) On multiple benchmarks, the proposed general-purpose two-stage inference framework can achieve a performance very close to (high-quality) large teacher’s performance by delegating a small fraction of the instances to the teacher, it clearly shows the utility of the framework.
>
> __Thus, the motivation of the proposed method is quite unclear.__
>
> We specified the motivation in Section 1, 4, and 5, which was appreciated by the other reviewers. To reiterate: Our motivation is to realize high performance at low inference cost by processing easy (and frequent) instances at a lightweight student model while invoking a large high-quality teacher model on hard (but rare) instances. This motivation is repeatedly stated in the paper (which according to the reviewer is fairly well written and easy to follow). We also note that our empirical results clearly establish that the proposed method/framework indeed demonstrates this desired behavior.
>
> __In addition, since the focus of this paper is on inference efficiency, it is necessary to compare the inference time, in addition to the current accuracy.__
>
> This is a good point. However, we would like to bring it to the reviewer’s attention that we had already included results for inference time/total latency in our original submission. Please see Figure 2c, 3c, and 8.

---

### Official Review · Reviewer_sKmp · 2021-10-31

**Correctness:** 3
**Technical Novelty And Significance:** 3
**Empirical Novelty And Significance:** 3
**Recommendation:** 6
**Confidence:** 3

**Main Review:**

Strengths
- It is a well written and clear paper providing a two-stage inference framework that can be used to reduce the dependency on large models and thus improve inference efficiency.
- The relevance of the framework is validated empirically on benchmark datasets in Computer Vision and NLP.

Weaknesses
- Even though the framework is clear and easy to use. The results are not very convincing as the performance of the two stage framework is very close to distillation which is also efficient and does not rely on the teacher model during inference. In particular in Figure 2, 3 and 4 curves of the proposed approach seem to be very close to the baseline standard distillation. The gains are clearer in Table 1 and Table 4. Could the authors add a similar table for Figure 4?
- It is not clear which approach (CD-I, CD-II, CD-III) and which parameters (alpha) to use in practice. Could the authors provide a summary of the findings regarding recommendations on which approach to use in which cases based on the experiences?

Questions
- In 4.1, the authors mention that « As a result of standard distillation, the lite student behaves as a well-calibrated model ». Could the authors justify this statement? I am not sure to understand why it is necessarily the case.
- In Table 1, the CD-II approach obtain a much lower Accuracy score than the other approaches (0.24), Could the authors explain that?

Typos
- Introduction, third paragraph : « real-world data »
- 3.2, 2nd paragraph : « the teacher scores »


**Summary Of The Paper:**

The paper proposes a two-stage distillation framework to improve inference efficiency and reduce the dependency on large teacher models. The goal of this framework is to only use the large/teacher model for difficult and rare examples and to use the student, smaller model for the more frequent easy examples. The procedure is composed of a training phase and an inference phase. During the training phase, the dataset is separated into two subsets: one containing the hard/difficult examples and one containing the easy examples. Using a loss incorporating a combination of label-smoothing and distillation, the student model is taught to be certain on the easy examples (by using distillation of the large model) and to be less certain on the hard/difficult examples. During the inference phase, in order to route the hard examples to the larger teacher model, several possible methods are proposed relying in particular on whether the margin of the student’s softmax is higher or lower than a threshold.

The authors validate their two-stage framework empirically on three benchmark image datasets (CIFAR-100, ImageNet-1k and ImageNet-21k) and two benchmark NLP datasets (SQuAD and MNLI).

**Summary Of The Review:**

Even though the performance of the two-stage framework seems to be higher than the standard distillation, I am not sure yet if the gains of the proposed approach compared to standard distillation justify the acceptance of this paper.

---

> ### Author Response · Authors · 2021-11-19
> **Response to Reviewer sKmp**
>
> We are happy that the reviewer found the paper clear and well written. Below we provide a pointwise response to your questions/concerns.
>
> __The results are not very convincing as the performance of the two stage framework is very close to distillation which…does not rely on the teacher model during inference…__
>
> We believe there is a misunderstanding here regarding the __Baseline__. Note that __all plots__ in Fig 2, 3, and 4 correspond to a two-stage inference where both teacher and student are utilized. The two extreme points of these plots correspond to using only teacher and student, respectively, during the inference. For example, the rightmost points in all the plots in Figure 2a (i.e., fraction processed by student = 1.0)  correspond to using only the student and the leftmost points in all the plots correspond to using only the teacher during inference. We had hoped to make it clear in Sec 5.1:
>
> “...Besides, we also include conventional model distillation as Baseline in our evaluation. Note that, for all approaches, allowing the lite student model to delegate hard instances…”
>
> We plan to make this point clearer to avoid any further misunderstanding.
>
> Key takeaway of Fig 2, 3, and 4 is that the distillation-based two-stage inference in itself (*with standard distillation or with proposed specialized distillation*) is powerful if one wants to approximate the performance of a teacher by only routing rare hard instances to the teacher. Additionally, Fig 2b, 3b, 4b and 4d show that if one expects the test instances to be imbalanced in favor of *in-domain* instance (e.g., instances from popular classes during inference) and care more about realizing higher performance gain on these *in-domain* instances, then two-stage inference based on specialized distillation can realize further gains over two-stage inference based on standard distillation (i.e., baseline).
>
> Note that two-stage inference based on standard distillation is also an instantiation of our generic distillation-based two-stage inference framework, which, to the best of our knowledge, has not been previously proposed.
>
> __Adding a table for Figure 4__
>
> Note  that Table 1 and 4 correspond to __class-based delegation__ whereas Fig 2, 3, and 4 employ __margin-based delegation__. As the caption of Table 1 states, unlike margin-based delegation, for given teacher and student models, we obtain a single value of (Accuracy, Fraction) tuple with class-based delegation. For margin-based delegation one obtains different (Accuracy, Fraction) tuples as one varies the delegation threshold $\rho$ in (7). Therefore, we have chosen to present the results for margin-based delegation as plots in Fig 2, 3, and 4 to make our presentation more concise and clearer.
>
> __It is not clear which approach (CD-I, CD-II, CD-III)...__
>
> Both CD-II and CD-III train a student with only popular classes as the potential output space (plus a rejection class in the case of CD-III). Thus, the student can not correctly classify the examples from non-popular classes. Thus, CD-I constitutes the preferable approach if we do want a student that has good overall performance (for all classes). In addition, CD-I also ensures good in-domain (i.e., only those examples that belong to popular classes) performance for a suitable value of $\alpha$. The exact value of $\alpha$ would depend on the underlying dataset and the number of popular classes (see, e.g., Fig 2b and 3b). For CD-I, the value of $\alpha$ is the key parameter that dictates the tradeoff between overall performance vs. in-domain performance. In particular, one can work with higher $\alpha$ to improve in-domain performance at the cost of degradation in overall performance.
>
> We will include a discussion along these lines to the revised paper.
>
> __Justification for the statement about “As a result of standard distillation…”__
>
> Note that the distillation process involves training a student with soft label distribution provided by a teacher as opposed to training with one-hot distribution. Since the soft label distribution provided by the teacher is well calibrated (with appropriate temperature scaling [Guo et al.]), in effect, the resulting student also behaves as a well-calibrated model.
>
> *[Guo et al.] On calibration of modern neural networks, arXiv:1706.04599.*
>
> __In Table 1, the CD-II approach obtain a much lower Accuracy score__
>
> Note that Table 1 employs class-based delegation. Since CD-II trains a student with only L’=30 classes (Sec A.1), it cannot delegate any examples to the teacher during inference. This leads to Fraction = 1.0, i.e., the student makes the final prediction on the examples from all 100 classes. Since the student can only hope to correctly classify examples from L’ = 30 classes, it makes errors on all the examples from the remaining L - L’ = 70 classes. In contrast, CD-III has an additional rejection class (Sec A.2) which allows some examples from L - L’ = 70 classes to be routed to the teacher.

---

> > ### Comment · Reviewer_sKmp · 2021-11-28
> > **Response Read**
> >
> > Thanks for the response. I am satisfied with the authors' response and I change my score from 5 to 6.

---

### Official Review · Reviewer_wETK · 2021-11-02

**Correctness:** 3
**Technical Novelty And Significance:** 2
**Empirical Novelty And Significance:** 2
**Recommendation:** 5
**Confidence:** 3

**Main Review:**

The paper tackles an important and practical problem: So-called "edge" devices such as mobile phones cannot process heavy duty models. However, requirements on bandwidth and latency may prevent one from sending every example to the cloud for processing on a big GPU capable machine. So, this paper presents a hybrid approach where most of the cases can be processed on the "edge" but those edge devices can elect to send rare cases to the cloud for a heavy duty model.

Despite being an important problem, the solutions in the paper "label delegation" and "margin delegation" are very heuristic and arbitrary. Solutions like early-exit networks seem much more principled in comparison. They are also crucially dependent on hyper-parameters, which the paper does not address tuning.

Heuristic solutions are still noteworthy for publication if they beat all other known approaches in a rigorous benchmark. However, in this paper, most of the solutions benchmark the method against variations on itself.

The paper would be stronger if it had more of these comparisons.

**Summary Of The Paper:**

This paper presents a new method to do efficient inference with a teacher-student setup for model distillation. It contrasts itself to traditional model distillation because it is optimized for the situation where rare and difficult cases can still be sent to the original teacher network.

**Summary Of The Review:**

The paper attempts to solve and important problem, but is very heuristic and does not validate that it is better than the prior art.

---

> ### Author Response · Authors · 2021-11-19
> **Response to Reviewer wETK**
>
>
> We thank the reviewer for taking the time to review our paper. We are glad that the reviewer recognized the importance of the problem considered in the paper. Before providing a point-wise response to the reviewer’s comments, we would like to reiterate the main contributions of the paper:
>
> 1) We propose a novel distillation-based two-stage inference framework that aims to utilize a lightweight student to make predictions on easy (but frequent) instances and delegate hard (but rare) instances to a large teacher.
>
> 2) We proposed novel distillation losses that aim to aid the student in routing hard instances to the teacher (based on a suitable delegation strategy). The novel loss functions focus on two specific instantiations of the generic two-stage inference framework: i) class-based distillation and ii) margin-based distillation.
>
> 3) We evaluate the proposed two-stage inference framework on both image classification and NLP benchmarks. Our experiments indeed show that:
>
>   a) Distillation-based two stage inference enables us to realize a desirable __inference cost vs. performance trade-off__ where we achieve performance comparable to the large teacher model while still processing a significant amount of test instances at the lightweight student.
>
>   b) When focusing on the *in-domain* performance (which corresponds to the instances that one expects to be more prevalent during the inference), the novel distillation losses further improve the performance of the two-stage inference framework for suitable label-smoothing parameter ($\alpha$).
>
> __“Label delegation” and “Margin delegation” are very heuristic…early-exit networks seem much more principled in comparison.__
>
> Note that “class (label) based delegation” arises as a natural delegation criterion in our class-based distillation procedure that already assumes some prior knowledge regarding popular vs. non-popular classes during inference. In contrast, “margin-based delegation” is motivated by the literature that uses margin as a predictor of the hardness of examples as perceived by the model, see, e.g., [Scheffer et al. 2001, Liu et al. 2019].
> We do not see it as apparent that early-exit networks are much more principled. Even such networks need to rely on an “exit criterion” at early layers and prior works often relied on margin (or closely related concepts such as prediction entropy and confidence) as the exit criterion, see e.g., [Schwartz et al. 2020, Xin et al. 2020].
>
> Please also see our later comment on early-exit networks vs. our proposal.
>
> *[Scheffer et al. 2001] Active Hidden Markov Models for Information Extraction*
>
> *[Liu et al. 2019] AdaptiveFace: Adaptive Margin and Sampling for Face Recognition, ICCV 2019.*
>
> *[Schwartz et al. 2020] The Right Tool for the Job: Matching Model and Instance Complexities, https://arxiv.org/pdf/2004.07453.pdf*
>
> *[Xin et al. 2020] DeeBERT: Dynamic Early Exiting for Accelerating BERT Inference, https://arxiv.org/pdf/2004.12993.pdf*
>
> __They crucially depend on hyper-parameters, which the paper does not address tuning.__
>
> We would like to highlight that our empirical evaluation in Sec 5 and appendix already presents results that involve tuning different hyperparameters. For example, Fig. 2, 3, 5 and Table 1 shows how varying the hyperparameter $\alpha$ affects the performance of class-based distillation. As for the delegation parameter $\rho$, varying $\rho$ changes the fraction of examples processed at the student in all the plots in Sec 5 and appendix.
>
> We would be happy to revise the text to *highlight* this exploration wrt. different hyperparameters.
>
> __In this paper, most of the solutions benchmark the method against variations on itself.__
>
> Note that our main objective is to approximate the performance of the high-performing large model while invoking the large model as few many times as possible. Thus, the performance of the (SoTA) large teacher serves as the clear benchmark to evaluate our solution against. Our experimentals indeed show that our novel distillation-based two-stage inference successfully achieves this desired objective on both image classification and NLP benchmarks.
>
> As for the existing work on reducing inference cost, as discussed in Sec 2, quantization and pruning is complementary to our approach and can be combined with our distillation framework. As for the works on adaptive computation (including *early-exit networks*), these methods require designing the whole training pipeline from scratch. In contrast, our framework *does not modify* the large model in any manner (because it's very expensive to train or modify large models and/or for some large models we only have API access) and only trains a lite student. Furthermore, unlike prior work on adaptive computation, our proposed framework is not tied to specific tasks and architectures. Thus, we believe that comparison to adaptive computation strategies is out of scope of this paper.

---

### Official Review · Reviewer_nhWf · 2021-11-02

**Correctness:** 3
**Technical Novelty And Significance:** 2
**Empirical Novelty And Significance:** 3
**Recommendation:** 6
**Confidence:** 4

**Main Review:**

Traditionally model distillation usually sacrifice performance for improved efficiency. In this paper, the authors proposed a two stage framework based on distillation which can achieve both the modeling benefits of the large models and preserve the efficiency with lightweight models. The key component is a well designed loss function for the student model which can help in the delegation process. Overall, I believe the paper is well motivated, the method is described clearly, and thorough experiments are conducted.

**Summary Of The Paper:**

This paper proposes a two stage distillation framework. By falling back hard samples to the large models and use small models for easy ones, such an approach could achieve both efficiency and performance.

**Summary Of The Review:**

I only have one major concerns towards the experiment. The authors propose several different approaches in the paper, including class-specific distillation and margin based distillation, and corresponding delegation methods. However in the experiment section it looks like they only focus on class-specific distillation and some related methods which are introduced in the appendix.

I would actually suggest the authors to make the paper more self-contained, if the methods are introduced in the main text they should be evaluated as well. And some alternative methods can be included in the appendix. Also, I feel like it would be better to have a high level summary in terms of the pros and cons for each method, along with the recommendation for users in practice.

Minor issues: line 4 of Introduction, upto -> up tp

---

> ### Author Response · Authors · 2021-11-19
> **Response to Reviewer nhWf**
>
> Thank you for your detailed comments. We are happy that the reviewer found our paper well motivated and noted the utility of the proposed two-stage framework based on the distillation in realizing modeling benefits of large models while ensuring efficiency with a lightweight model.
>
> __Experiments on margin-based distillation__
>
> We would like to note that we *do* conduct experiments for the margin-based distillation. In particular, the results on CIFAR-100 and ImageNet-1k are presented in the Appendix E. We highlight this at the end of Section 5.2: *“Finally, we also studied the in-domain performance of two-stage inference enabled by margin-based distillation from Sec. 4.2 on both CIFAR-100 and ImageNet-1k in the appendix.”*
>
> Note that our results on MNLI (Figure 4a-4b) are also based on margin-based distillation. Furthermore, the results on SQuAD (Figure 4c-4d) are based on a suitable modification of margin-based distillation. Please see Section 5.3 for the details.
>
> __Making paper more self-contained__
>
> We thank the reviewer for the suggestion. As mentioned above, the paper does include experimental results for all the methods introduced in the main text and appendix. That said, we would revise the text to ensure that all such results are more clearly highlighted. In addition, we will include additional discussion on the pros and cons of different methods covered in the paper.
>
> We have fixed the typos pointed out by the reviewer.

---

> > ### Comment · Reviewer_nhWf · 2021-11-30
> > **Thanks for the response**
> >
> > Thanks for the response to address my concerns.

---

### Decision · Program_Chairs · 2022-01-20

**Decision:**

Reject

**Comment:**

This paper a distillation framework where a light-weight student model is trained to handle easy (frequent) instances, while the large teacher model is still used to handle the more difficult (rare) inputs. The models are trained to perform well in this two-stage inference setting. Experiments are conducted on computer vision and NLP tasks. While the idea is potentially interesting, the experimental results are fairly weak and not very convincing.